# Microbial Control of Raw and Cold-Smoked Atlantic Salmon (*Salmo salar*) through a Microwave Plasma Treatment

**DOI:** 10.3390/foods11213356

**Published:** 2022-10-25

**Authors:** Thomas Weihe, Robert Wagner, Uta Schnabel, Mathias Andrasch, Yukun Su, Jörg Stachowiak, Heinz Jörg Noll, Jörg Ehlbeck

**Affiliations:** 1Leibniz-Institute for Plasma Science and Technology, 17489 Greifswald, Germany; 2Naval School of Technology, 18445 Kramerhof, Germany; 3Institute for Sports Science, University of Rostock, 18051 Rostock, Germany; 4Feinfischräucherei Noll, 46514 Schermbeck, Germany

**Keywords:** seafood pathogens, food sanitation, food spoilage, microbiological inactivation, microbial growth, food quality, shelf-life prolongation, quasi-thermal plasma, reactive nitrogen species

## Abstract

The control of the pathogenic load on foodstuffs is a key element in food safety. Particularly, seafood such as cold-smoked salmon is threatened by pathogens such as *Salmonella* sp. or *Listeria monocytogenes*. Despite strict existing hygiene procedures, the production industry constantly demands novel, reliable methods for microbial decontamination. Against that background, a microwave plasma-based decontamination technique via plasma-processed air (PPA) is presented. Thereby, the samples undergo two treatment steps, a pre-treatment step where PPA is produced when compressed air flows over a plasma torch, and a post-treatment step where the PPA acts on the samples. This publication embraces experiments that compare the total viable count (tvc) of bacteria found on PPA-treated raw (rs) and cold-smoked salmon (css) samples and their references. The tvc over the storage time is evaluated using a logistic growth model that reveals a PPA sensitivity for raw salmon (rs). A shelf-life prolongation of two days is determined. When cold-smoked salmon (css) is PPA-treated, the treatment reveals no further impact. When PPA-treated raw salmon (rs) is compared with PPA-untreated cold-smoked salmon (css), the PPA treatment appears as reliable as the cold-smoking process and retards the growth of cultivable bacteria in the same manner. The experiments are flanked by quality measurements such as color and texture measurements before and after the PPA treatment. Salmon samples, which undergo an overtreatment, solely show light changes such as a whitish surface flocculation. A relatively mild treatment as applied in the storage experiments has no further detected impact on the fish matrix.

## 1. Introduction

Hygiene is a significant element in the food production chain, i.e., the outgoing produce has to be within microbial specifications given by a national or international legislator [1,2,3]. Given this background, poor-hygiene production environments and inadequate sanitation will result in healthcare-associated infections and foodborne diseases as well as high production losses of food [4,5]. In particular, foods such as fish, seafood, and freshly cut leafy greens embrace a group of produce that are sensitive to handling and vulnerable to microbial infections in the whole value chain [6]. Despite the wide variety of potential human pathogens, the prevalence of the foodborne pathogen *L. monocytogenes* on seafood has dramatically increased [7]. For instance, in 2019 the most-sampled RTE (ready-to-eat) food categories of all reported member states for an *L. monocytogenes* contamination embraced 6.1% RTE fish and fishery products such as cold-smoked salmon [8]. A paper of Rørvik et al. [9] described sub-clones of *L. monocytogenes* found in hospitalized patients with an origin in smoked salmon. On the other hand, *L. monocytogenes* may persist in food production facilities and cause repeated, seemingly sporadic illnesses over extended periods of time [10]. Beside hazardous outbreaks of listeriosis and their severe consequences for consumers [10], salmonellosis is still a very common gastrointestinal infection in humans [8,11,12,13]. As early as 2016, Friesema et al. [14] described a severe outbreak of *Salmonella enterica* subsp. enterica serovar Thompson, also found in samples of smoked salmon in the Netherlands, which led to the hospitalization of 35% of all infected patients. Throughout Europe, 94,625 confirmed salmonellosis cases have appeared, which is a notification rate of 21.2 cases per 100,000 population. Both pathogens infest predominantly the elderly and immune-suppressed.

Smoking has been traditionally applied as a preservation method, but the gustatory influence of smoking nowadays tends to attract focus [15]. Furthermore, smoking is used primarily for its sensory advantages such as an appealing color or taste in minimally processed products with a lower salt content [16,17]. The smoking of salmon in particular, which can embrace cold smoking at temperatures below 30 °C or hot smoking at high temperatures above 60 °C, delivers a distinctive taste. The process starts with the pyrolysis of wood to polymers such as cellulose, hemicellulose, or lignin. Generally, wood smoke consists of light-scattering tarry droplets suspended in a medium of air and invisible vapors, which contains many various compounds such as aldehydes, ketones, alcohols, acids, hydrocarbons, esters, phenols, and ethers. Due to an incomplete burning of the wood, compounds such as phenols, aliphatic and other aromatic compounds, carbonic acids, and the cariogenic benzopyrene also form smoke ingredients [16,17,18]. Virtually all of them are antimicrobial-effective [17,19]. Several authors describe the deposition of smoke as a mixture of tarry nanoparticles and molecular smoke by chemical reactions with water. Both parts are different in their chemistry. For instance, wet produce is found to be more prone to the smoking process [20]. Although preserving foodstuffs by smoke curing has been performed since prehistoric times, it is only in the past several decades that smoking has been studied in detail [20,21].

Given this background, the presented plasma-based decontamination technology complements the traditional smoking technique for preservation. Much like the smoking process that retards pathogen growth with chemically altered compounds of wood smoke, the gaseous plasma-based technique wins its preservative function through an energetic alteration of air compounds such as nitrous gases. In various studies, PPA has already been used for the decontamination of food. For instance, Durek et al. described the impact of PPA on microbial count and their diversity found on dried herbs in a pilot-scale experiment [22]. The same working group also examined the microbiological inactivation of PPA in fresh pork under the prerequisite of maintaining produce quality [23]. The exact mechanism of action of PPA is currently under investigation. However, several authors predominantly address that antimicrobial power to nitrous gases that develop due to the plasma-generated chemical species [24,25,26].

Here, we present an innovative food sanitation method for cold-smoked foodstuff products such as smoked salmon. The publication presents a study that compares raw salmon (rs) and cold-smoked salmon (css) with PPA-processed samples of raw salmon (rs) and cold-smoked salmon (css). We conduct physical, chemical, and microbiological experiments to determine the impact of PPA on the growth of the natural microflora with a distinct focus on Salmonella and Listeria. These investigations are underpinned by characterizations of the development of the samples’ color and firmness against the background of the PPA treatment.

## 2. Materials and Methods

### 2.1. Sample Preparation

The samples included in the experiments consist of raw (rs-sample) and css sample Atlantic salmon (*Salmo salar*). The salmon are cut into halves, disemboweled, and smoked or flavored at the smoke house “Feinfischräucherei Noll” (Schermbeck, North-Rhine Westphalia, Germany). Subsequently, the fishes are packed, deep-frozen, and shipped overnight to the “Leibniz-Institute for Plasma Science and Technology” (Greifswald, Mecklenburg-Western Pomerania, Germany). Before every experimental set, the halves are thawed overnight in a refrigerator at 4 °C. For sample preparation, they are cut into cubic specimens of an edge length of 10 mm and a weight of approx. 10 g. To avoid any evitable decontamination, all steps for sample preparation are carried out under aseptic conditions.

### 2.2. Plasma Treatment

The samples are treated in a single-stage atmospheric plasma source used throughout the experiments. It is a microwave-driven (2.45 GHz) pulsed plasma generator, which has a pulse-period rate (on/off) of a period length of 10 s, where a 5 s plasma on-phase is implemented. Consequently, every 5 s on-phase is followed by a 5 s off-phase. It has a power injection of approx. 1.1 kW. Accordingly, the gas temperature is about 4000 K using a gas flux of 18 slm air as a process gas. The gas generator is described in greater detail in the publication of Andrasch et al. [27].

In principle, the plasma treatment subdivides into two steps. First, PPA is produced with technical dry compressed air, which is carried over the plasma torch. Subsequently, the gaseous PPA enters a washing bottle that increases the PPA’s bactericidal power. This pre-treatment (PT) leads to gaseous PPA that gathers in a glass bottle (3.3 borosilicate, Carl Roth, Karlsruhe, Germany). In the presented experiments, three different timeframes for a PT are applied (5 s, 15 s, and 50 s), whereas the dimension of the timeframes only reflects the ignition of the plasma torch. For instance, a 5 s pre-treatment embraces a whole pulse period (which has a duration of 10 s), i.e., a 50 s PT includes 5 periods and a total plasma on-time of 25 s. The second step embraces the antimicrobial treatment; the freshly generated PPA flows over a conduit system into a decontamination chamber where the rs and css samples have been placed. The so-called post-treatment (POT) has timeframes of 1 min, 3 min, and 5 min length. Since the PPA flows over high-grade steel pipes into the treatment chamber, it reaches the chamber at a temperature of approximately 24 °C, which is approx. at the same temperature that the smoke gasses in the cold smoking process.

### 2.3. Microbiological Experiments

#### 2.3.1. Storage Tests

The underlying microbiological data for all samples are determined in storage tests (*N* = 5). The storage tests are conducted in a stainless steel refrigerator (Profi 700 GN 2/1 static cooling, GastroHero, Dortmund, Germany) at a temperature of 4 °C for 21 days. This experimental setup describes a general modus operandi, unless the samples undergo a massive bloom of the ongrowing microflora, which has been observed for samples of the PPA treatment. However, those samples did not undergo the 21-day storage period. For convenience, the experiments are stopped after 14 days and the microflora’s tvc are determined. Generally, the samples are stored in polypropylene containers (GLIS, L × B × H: 17 cm × 10 cm × 8 cm, IKEA, Sweden) in the refrigerator immediately after their preparation. Samples undergoing a plasma treatment are briefly stored in the immediate vicinity of the plasma source. To ensure a uniform treatment scheme, the references embraced by that experiment are also stored at the plasma source in the same manner. The references complete the experimental setup and are stored in a refrigerator at 4 °C after each treatment. The microbial load is determined at the beginning of the storage and after the treatment (day 0), on the 7th, 10th, and 14th day.

#### 2.3.2. Recovery, Detection, and Evaluation of Native Contamination

The quantification of the residual microflora after a PPA treatment is revealed by a homogenization of the samples, which are subsequently suspended, diluted, and plated on agar plates (plate-count agar (PCA), sifin diagnostics GmbH, Berlin, Germany). The PCA plates are cultured for the amplification of various endemic species at 33 °C for 24 h. Subsequently, the experimental evaluation of the plates serves the bacterial load in the decadic logarithm (log_10_) of the total viable count (tvc) in colony-forming units per mL (cfu/mL). Therefore, a single sample is transferred into a homogenizer bag complemented by 10 mL of a homogenization solution (phosphate-buffered solution (PBS) at pH 7.2 after Sörensen). Subsequently, the mixture is homogenized for approx. 1 min at a speed of 260 rpm (SewardTM StomacherTM Model 400C Circulator Lab Blender, Seward Limited, Worthing, West Sussex, United Kingdom). The suspension is used to make a serial dilution (1:10,000–1:10,000,000,000) that is transferred onto the agar plates. As mentioned above, the data points embrace five repetitions under the same conditions (*N* = 5).

### 2.4. Quality Control

To assess the impact of the PPA treatment on the fishy matrix of the samples, various measurements are carried out. The aim of these experiments is to achieve widely comparable data that are captured by a reliable technique. To that end, some modifications of the data evaluation are necessary. For instance, the distances in the color space between two stimuli are not presented in C-values as it is suggested by Pathare et al. [28].

#### 2.4.1. Pictures

All pictures are taken with a commercially available camera (Lumix DMC-LX7; Panasonic Cooperation, Kadoma, Japan). This work solely presents these pictures as a coarse-grained analysis that mirrors possible uncertainties and variances on the sample surface that might not be captured methodically.

#### 2.4.2. Texture Analysis

A texture-analyzing device (TAXT+, Stable Micro Systems Ltd., Surry, UK) is used to determine potential PPA treatment-dependent texture changes of the treated samples. The measurements are carried out directly after each PPA treatment (*N* = 5) and are compared with an untreated reference (*N* = 5). In those measurements, force is applied normally to a treated sample surface and its references. The experimental design simulates a bite at a distinct force, which mirrors the consistency of the sample matrix. The results are displayed in N (force). The samples, placed in a beaker underneath the texture gauge, are measured with a measuring head made from a plastic blade that depicts a row of human teeth (Stable Micro Systems, Light Blade Knife (A/LKB), Godalming, UK). The measurements reflect the forces that dissect the sample matrix into two halves. The results are compared based on a t-test. Each sample (10 g) is PPA-treated according to the parameters mentioned earlier (PT: 5 s, 15 s, and 50 s; POT: 1 min, 3 min, and 5 min). The samples are placed on a paper towel to contain excess water.

#### 2.4.3. Color Stimulus Specification

The color stimulus analysis (*N* = 3) is carried out in the Hunter Lab-system and the values of L (lightness), a (redness), and b (yellowness) are determined. The measurements are carried out with a portable colorimeter (NH310, 3nh, Shanghai, China) under standardized measurement conditions such as illumination and a 0° observation angle. ΔE (DIN EN ISO 11664-4) as a single-valued indicator for the total color difference between PPA-treated rs and css samples and their untreated references is determined as follows:(1)$$∆E=\sqrt{{\left(∆L_{S}-∆L_{0}\right)}^{2}+{\left(∆a_{S}-∆a_{0}\right)}^{2}+{\left(∆b_{S}-∆b_{0}\right)}^{2}}$$
where *L_S_*, *a_S_*, and *b_S_* are the color values of the PPA-treated rs and css samples and *L_0_*, *a_0_*, and *b_0_* are those for their untreated counterparts directly after thawing.

### 2.5. Statistics

#### 2.5.1. General Statistics

According to the work of Limpert [29], all values obtained from microbiological experiments are presented as the decadic logarithm (log_10_) of the cfu-counts, which have been determined on agar plates as described above. Consequently, the data distribution is assumed as a right-skewed distribution function, which is straightened as a log-normal distribution for convenience. The assumption is underpinned by a series of Shapiro–Wilk tests (α = 0.05, df = 4 (microbiology, texture), 2 (color)), which prove normality throughout for the observed data. Additionally, all data are checked for variance homogeneity by a Levene test (α = 0.05, df = 4 (microbiology, texture), 2 (color)).

Thus, for further calculations such as a model calculation, calculation of the arithmetic mean, or error propagation, all microbiological values are treated as normally distributed, which opens the possibility to use them in parametric statistical tests. The findings are mirrored by a logistic regression using a simple logistic growth model for a single bacterium. They are underpinned by a multivariate analysis of variance (MANOVA), where the tvc found on day 7 and day 14 of the storage period is compared for every treatment group (5 s, 15 s, and 50 s). We set the significance level to α = 0.05, but stronger statistics are reported, too.

Due to variance inhomogeneity, the data for texture measurements are evaluated by a robust MANOVA. An equivalence test for the measurements of the color stimulus specification does not work due to the non-normal data distribution (R-project, multeq.diff). A robust MANOVA for normally distributed values out of the data set (reference vs. PT/POT 50 s/5 min) is conducted. We set the significance level to α = 0.05, but stronger statistics are reported, too. For data sets not normally distributed, the median instead of the arithmetic mean is taken into account.

#### 2.5.2. Model

We choose a logistic growth model to describe a beginning ongrowing microbial load on rs and css in a storage experiment over 14 days. A logistic growth function [30] is fitted in an iterative routine to the data *P(t) = log_10_(cfu)*,
(2)$$P\left(t\right)=\frac{K}{1+e^{\left(-\alpha \left(\beta -t\right)\right)}}$$
where *t* is the time point of sampling, α [1/d] mirrors a growth-rate constant of a natural, sample-specific microbial flora that is detected on the salmon halves. Furthermore, β [d] is a somehow triggered beginning of the start-up phase (end of the lag-phase) for an exponential microbial growth, and κ [log_10_(cfu)] a population limit in terms of a load capacity. The data are fitted using a non-linear regression model [31]. The selected curves are determined by the maximum likelihood method.

## 3. Results

### 3.1. Raw and Smoked Salmon vs. Their PPA-Treated Counterparts

Figure 1a,b merges the data of the tvc, which were experimentally obtained both for untreated/PPA-treated rs (Figure 1a) and untreated/PPA-treated (Figure 1b) css. Both figures also include the logistical regressions based on these data. In contrast to prior experiments that solely compared the tvc of css and rs over a storage period of 21 days, the maximum microbial load (approx. log_10_ = 8.0 cfu/mL) appeared in a shorter storage period of 14 days for all rs (Figure 1a) and css samples (Figure 1b). During the PPA-treatment, the all-salmon samples, which also enclosed the references, were transferred to the treatment site close to the plasma source. The samples were stored in the refrigerator at 4 °C when no experiments were carried out. In these experiments, the rs samples and the css counterparts were both stored for 14 days. Before storage, the samples underwent various PPA treatments with different intensities (PT: 5 s, 15 s, 50 s), which all acted upon the samples for a POT of 5 min.

Particularly, Figure 1a reveals significant differences in its array of curves when compared with Figure 1b. The α-parameters, which were obtained by logistic regression, are not very different when individually compared with the α-value of the untreated reference (maximum 0.54 ± 0.27 for a PT of 5 s and a minimum of 0.38 ± 0.26). Among the array of curves for rs, the curve for a PT of 15 s is an outstanding result due to its very low α value, which flattens the curve potentially oversized. In contrast, the β-values vary between PPA-treated and rs (minimum of 4.07 ± 0.36 days for a PT of 50 s, maximum of 5.13 ± 1.01 days for a PT of 5 s) and its PPA-untreated rs reference (2.79 ± 0.36 days). The maximum, microbial load capacity (κ) is additionally significantly higher for PPA-untreated rs samples (tvc of log_10_(cfu/mL) 9.02 ± 1.29) than for their treated counterparts (reference rs, tvc of log_10_(cfu/mL) 7.68 ± 0.30 after a PT of 50 s and max–7.94 ± 0.36 for a PT of 15 s).

However, PPA-treated and untreated css samples do not show any significant difference for the parameters and all parameters differ in a narrow range. The css samples, extraneous if PPA-treated or not, reveal very similar values for their logistic regression.

These findings, obtained from the logistic regression, were underpinned by a multivariate analysis of variances (MANOVA) that compared the tvc on the 7th day and the 14th day of all PTs followed by a 5 min POT. In both cases, the tvc significantly rose over the storage period as it is described above (rs: *p* < 0.002, css: *p* < 0.02).

There is no meaningful statistical difference between the pre-treatments and the untreated reference when css is used (*p* > 0.05). The rs shows a statistical meaningful reaction on a plasma treatment in general (*p* < 0.05), but no significant increase in bactericidal power by a longer PT could be obtained (*p* > 0.05). For a POT of 1 min and 3 min, the MANOVA detected no meaningful statistical variation in the tvc due to a plasma treatment. Since milder PPA treatments show no distinct effect on salmon samples, the data have not been presented within the scope of this publication. However, the approach to these types of experiments was the same as those listed here.

### 3.2. Quality Control

Figure 2 generically shows a photograph of PPA-treated rs at optimum conditions (Pre: 15 s and POT of 5 min) versus a PPA-untreated reference (left hand side in every individual picture of Figure 2). The samples were observed for a 24 h period after the treatment. In comparison with their references, the color of the samples did not remarkably change in local inspection due to a PPA treatment, unless treatment parameters such as a PT of 50 s and a POT of 5 min were deployed. After such a treatment, obvious changes such as color alterations, flocculation of the salmon matrix (also impermanent and obvious for a PT/POT: 15 s and 5 min, Figure 2 red circle) compounds or a loss of water content frequently appeared (not shown). Alterations of the samples did not necessarily stop immediately after a PPA treatment but also became apparent for PPA-untreated references. Generally, color variations among all samples were widely observed. Photographs were taken for PPA-treated rs vs. a rs reference (PPA-untreated) and PPA-treated css vs. a css reference (PPA-untreated salmon) for all conducted PT and POT. All further investigations were based on color stimulus measurements on the Lab-System. The Lab-System depicts a certain color as a set of L-, a-, and b-values. The L-value depicts the brightness of the color, the a-value represents a color spectrum from green (-) to red (+), and the b-value represents an axis that covers the colors from blue (-) to yellow (+). All representations show values, which are taken after an idealized PPA treatment of a PT of 15 s and a POT of 5 min. Additionally, Figure 3 summarizes and compares the ΔE-values calculated from color stimulus measurements of PPA-treated rs and css samples (Equation (1)). The figure shows samples that have been treated with different PTs (5 s, 15 s, and 50 s) and POTs (1 min, 3 min, and 5 min).

Figure 3 compares both the total color difference ΔE for all samples and their naturally occurring color variations and color changes due to a PPA treatment. It is particularly striking that all graphs from A to F have their origin in the data point POT/ΔE = 0/0 for all PTs. For the determination of possible color changes after a PPA treatment, every PPA-treated sample is compared with itself before its treatment. Consequently, ΔE becomes zero when every rs and css sample is compared with itself at a treatment time point t = 0 s. The basic color values were determined directly after the treatment. For the determination of ΔE values for PPA-treated rs and css samples, the samples were observed after 24 h at different PTs (1 min, 3 min, and 5 min). Additionally, it is obvious that css samples do not show massive color alterations in terms of a high ΔE-value.

Figure 3A,D shows the untreated references for rs (D) and css (A) samples. For all references, the strongest color alterations were observed for the references that were used for comparison in the 5 s-PT experimental set (ΔE = 7.02 ± 2.21 after a POT of 1 min). The ΔE-values decreased for that experimental sequence. That means the strongest color changes were observed after the samples were in contact with ambient air. Please note the described samples did not undergo any PPA treatments. Nevertheless, no ΔE-values for their treated css counterparts (Figure 3B,C) showed higher total color changes (highest value for PPA-treated css samples: ΔE = 6.77 ± 2.42 for PT of 5 s after a POT of 3 min). Additionally, all values showed no general tendency, which might be interpreted as a direct function neither of the PT, the POT treatment time, nor of the storage period of 24 h. Equivalence tests failed because no normal distribution could be detected. Values, appearing in the graphs, are medians. However, based on a robust MANOVA, color changes or color difference may not be excluded among the reference groups (robust MANOVA, reference vs. PT/POT 50 s/5 min, *p* < 0.05).

Figure 3D–F represents the total color changes for the reference (D), the ΔE for treated rs samples after they underwent that regime (E), and the ΔE for PPA-treated rs samples after a 24 h storage period (F). They show a slightly different picture. For those samples, an increasing color difference for longer POT times might be an underlying process due to compounds found in the PPA (highest ΔE-values for rs-samples: 17.37 ± 4.85 for a PT of 15 s and a POT of 5 min). It is obvious that the lowest observed ΔE-value for rs samples is in the range of the highest value found for css samples (lowest ΔE for rs samples: 2.37 ± 2.44 for a PT of 5 s and a POT of 1 min). As it is highlighted above, color alterations due to the influence of a PPA treatment cannot be excluded (robust MANOVA, reference vs. PT/POT 50 s/5 min, *p* < 0.05).

The experiment mimics the so-called “first bit” and determines the force with which a tooth-imitating blade needs to dissect the fish specimens. Unless the samples are maximally treated (PT: 50 s, POT 5 min), the force appears slightly different for rs but not for css at any time point of the PT. Furthermore, it becomes obvious that smoked salmon does not have any major texture changes when a PT no longer than 15 s is applied. After a PT of 15 s, the firmness of the tissue slightly increases (css, PT 15 s: 46.73 N ± 14.66 N; css, PT 50 s: 67.35 N ±13.56 N). That effect appears vice versa for rs specimens. For PTs < 5 s, the firmness of the tissue increases and reaches a maximum force measured when the specimens undergo a PT of ≤15 s. However, either for rs or css, no variation appears significant in a robust MANOVA (robust MANOVA, rs vs. css, *p* > 0.05).

Figure 4 shows texture specifications for PPA-treated (POT of 5 min) rs - and css - samples compared to PPA-untreated references (referred as PT-treatment of 0s). The experiment mimics the so called “first bit” and determines the force, which a tooth-imitating blade needs to dissect the fish specimens. Unless the samples are maximal treated (PT: 50 s, POT 5 min), the force appears slightly different for rs but not for css at any time point of the PT. Furthermore, it becomes obvious; that smoked salmon does not have any major texture changes when a PT no longer than 15 s is applied. After a PT of 15 s the firmness of the tissue slightly increases (css, PT 15 s: 46.73 N ± 14.66 N; css, PT 50 s: 67.35 N ±13.56 N). That effect appears vice versa for rs - specimens. For PTs < 5 s, the firmness of the tissue increases and reach a maximum force measured when the specimens undergo a PT of ≤ 15 s. However, either for rs or css, no variation appears significant in a robust MANOVA (robust MANOVA, rs vs. css, p > 0.05).

## 4. Discussion

The experiments revealed the feasibility of a PPA treatment for raw and cold-smoked salmon. Therefore, rs and css samples were PPA-treated in a series of various experimental parameters such as PPA strength and treatment time. Subsequently, they were microbiologically compared with each other and their untreated counterparts in storage experiments. These investigations were accompanied by studies of the produce quality based on texture measurements and color comparisons. The experiments showed an antimicrobial effect of PPA treatment with a POT of 5 min on rs samples and thus could increase the shelf life of the samples by about two days (β-values, ref.: 2.79 vs. PT of 5 s: 5.13). However, our results showed that the PPA-based decontamination prolongation range is comparable with existing decontamination methods [32,33].

The data were evaluated by a logistic growth model. The regression model did not converge in every case (rs, PT: 50 s; css, PT: reference, 15 s), possibly due to an increased growth of bacteria entering the log phase late and interfering with the logistic model. Conclusively, the interpretation of the data presented utilizes a model for logistic growth as it is frequently described in the literature for a single microbe strain [34]. Contrary to the common use of such a model [35,36], it is used for a variety of microorganisms. However, for a detailed description of growth processes, various authors introduce variables and terms for its refinement. For instance, Gimenez et al. [36] offer various growth models for several microorganisms (*L. monocytogenes*, *Photobacterium phosphoreum*, *Enterobacteriaceae enterococci*, and various lactic acid bacteria). Further, spoilage primarily appears due to microbial activity and temperatures above 10 °C. Lactic acid bacteria dominate the spoilage microflora, sometimes together with *P. phosphoreum* or Enterobacteriaceae. They can reach their maximum population density rapidly and remain at that level during 50% of the product’s shelf life [7,37]. In the presented experiments, our mathematical model aimed to explain the development or the interplay of microorganisms composing the cultivable part of the microbiome found on css and rs samples.

Therefore, the findings of the logistical-regression model are supported by a multi-variate analysis of variances (MANOVA, SOT: 0 s, 5 s, 15 s, and 50 s vs. day 7 and day 14). All analyses show a decontamination of rs but no statistically meaningful effect on css. Against that background, the shelf life of vacuum-packed css predominantly varies according to the salt concentration and the storage temperature. In this regard, studies performed by various authors reveal a specific spoilage mechanism that differs from the rs spoilage [38,39,40]. Consequently, several authors mention a spoilage of css that does not necessarily correlate to the tvc found on rs [41,42]. This might explain the varied resilience of rs- or css-hosted microorganisms to a PPA treatment. In contrast, a comparison of untreated rs and css varied throughout the experiments. A clear antimicrobial effect of the cold smoking is noticeable, which is not detectable when rs and css references of the PPA treatment are compared. As fish is a highly vulnerable produce, contamination of the sample during the plasma treatment might be possible and should be considered when discussing experimental results. Additionally, all samples show a high variance, which might be due to differently contaminated parts of the samples.

To support the results of these studies, the experimental design should consider hygienic aspects. Nevertheless, the microflora of css host bacterial groups show a higher spoilage potential than others. For instance, Stohr et al. [41] identify bacteria mainly responsible for spoilage. The authors predominantly identify *Lactobacillus sake*, *Companilactobacillus farciminis*, and *Brochothrix thermosphacta* as spoilage-inducing bacteria that produce sulfurous, acidic, and rancid off-odors, respectively [41,43]. Both css and rs may also involve health risks bearded by the naturally occurred load, especially from the presence of *L. monocytogenes* [44,45]. Consequently, the microbiome of raw salmon may vary from those found on css, but bacteria such as *L. sake*, *B. thermosphacta,* or *L. monocytogenes* have also been found on raw salmon filets [39,46,47,48]. In summary, css and rs may naturally host a different composition of their microbiome but host similar spoilage bacteria and their pathogens, which conclusively determine their shelf life and do not alter their resilience to a PPA treatment in a pronounced way. Additionally, no statistically meaningful increase in decontamination power due to a stronger treatment was detected. As a stronger pre-treatment delivers a higher concentration of reactive oxygen and nitrogen species (RONS), thus gaining the bactericidal power of the PPA, all the treatments presented in this publication might appear as an over-treatment. However, a lighter treatment was not in the scope of our set-up. Additionally, stronger treatments may influence the underlying matrix and might support bacterial growth through an increase in the food supply due to damaged matrix cells. Additionally, a food analytics company (Eurofins Scientific) periodically checks the salmon used in the experiments in terms of their microbiological load. Consequently, the samples were delivered with a very low microbiological load (below the detection limit, Figure 1), and the observed growth of the microbiome is a result of the microbiological conditions on-site rather than an initial load of production-related microorganisms. However, the microbiome attenuated after a decontamination step such as a cold smoking or a PPA-treatment. Since the results of the PPA treatment align with those of a cold-smoking process, the PPA treatment appears as sufficient as a cold smoking. When the data were evaluated with the described model, all samples, which underwent either a cold smoking or a PPA-treatment, showed no significant variation in maximum load capacity (κ) or the time the concentration of cultivable bacteria reached κ/2 (β) (rs 5 s(β): 5.13 d, css ref (β): 4.86 d). Small variations of the microbiome´s composition may appear due to local variations in the environment of the fish [49] or due to various contamination of the production site [50,51]. Nevertheless, since the growth of the microbiomes of rs and css appear identically within their groups throughout the experiments, an identical vulnerability to PPA was assumed. For data evaluation, a lower load of microorganisms paired with possibly unchanged quality parameters were interpreted as a successful treatment, which is feasible for profitable use in the production industry. A common cold-smoking process already revealed a shelf-life extension of salmon (css) in preliminary tests, where compounds of condensed wood smoke acted antimicrobially [10,12,14,20]. The experiments with PPA revealed pronounced microbial properties on samples of raw and completely unprocessed salmon (rs samples). The css samples showed no further increase in their preservability in those storage experiments but the maximum antimicrobial effect on rs samples is comparable with that of a cold-smoking process.

Additionally, conventional plate count methods only provide information about the ability of bacteria to proliferate but no information is given about the physiological state of the bacteria. A lack of cultivability on culture media is not equivalent to cellular death because the bacteria can still be metabolically active causing diseases and reducing shelf live. The whole microbiome also embraces bacteria, which is due to inadequacies such as temperature, nutrient deficiencies, or lacking interactions to needed non-cultivable partners. Mechanical removal of bacteria is also possible when forces acting laterally on the sample surface randomly remove microorganisms. Additionally, chemical stress factors can feasibly force bacteria into a VBNC state. For instance, treatments with ammonium-based or peroxide-based disinfectants may induce an increase in VBNC populations of L. monocytogenes formed in smoked salmon processing environments [52]. Since the described PPA gas generator predominantly delivers RNS, an additional PPA treatment may influence the VBNC state for various bacteria [53].

Quality measurements flanked the experiments that determined the tvc of the treated samples and their references. All samples were photographed and were evaluated by a visual inspection. When the samples were treated with a relatively mild PPA gas (PT not longer than 15 s) no color changes or obvious alterations of the outer fish matrix were observed. All “mild”-treated (PT 5 s, POT 5 min) samples showed no significant quality loss. The appearance changed partly when PPA was used for an over-treatment. Here, whitish flocculation appeared upon the PPA-treated surface. We interpreted this flocculation as denaturized proteins hosted in the outer part of the fish matrix. Subsequently, all values of all samples and references were compared with a salmon color standard (SalmoFan) that specifies the desired color palette of raw and smoked salmon. The color stimulus specification also embraces a determination of the distances of the colors in the Lab-color space in terms of ΔE-values. Both rs and css showed color changes over a time period of 24 h; however, the color change was not associated solely with a PPA treatment since color changes were also observed for untreated references. Yet, color changes cannot be excluded completely due to the redox power of the RONS and other chemical compounds typically found in PPA. Although the light images of the samples at the strongest PPA treatment showed signs of an overtreatment, the color changes occurred in the range of a natural variation. Irregular white flake-like aggregates were observed on the surface of such specimens. Texture variations were statistically detected. For the texture analysis, no statistically significant changes in the firmness of the fish matrix were observed based on a robust MANOVA (*p* > 0.05). The texture analysis, which is subject to high standard deviations, showed an increased firmness of the sample matrix for raw salmon after a PPA treatment. This process was pronounced for PT of <15 s and ran into a plateau thereafter. After a PT of 50 s, the force values to dissect a PPA-treated raw salmon sample and a PPA-treated smoked sample were almost the same and did not differ significantly (smoked salmon: 72.78 N ± 10.67 N vs. raw salmon: 76.34 N ± 43.76 N). We interpreted that finding as a cooking-like procedure that became obvious for PPA post-treatments longer than 15 s. Besides the antimicrobial potential of the gas load of the salmon, these cooking-like behaviors may additionally act as a curing method.

In the food industry, pickling is an established curing method for the preservation of foods such as meat, cheese, and fish or seafood. Pickling is a conservation step through the addition of nitrous compounds such as curing salt [54,55]. The behavior of the nitrous gases and their further reactions to nitric and nitrous acid, which subsequently provide the antimicrobial potential, seem to be similar to ordinary curing as a process for food preservation. Consequently, a significant source of nitrate in the diet is in cured meat, in which it is used to impart color, flavor, and microbiological stability. Consequently, a PPA treatment where nitrate and nitrous compounds have been introduced upon the foodstuff surface leads to very comparable reactions counteracting the microorganisms.

Beside the conservation effect, nitrous compounds may develop hazardous compounds when used for conservation of food. For instance, N-nitrosodimethylamine appears to be a hazardous carcinogenic compound that establishes during the curing process [56]. Comparable to the salt-curing process, PPA treatments increase the concentration of nitrous compounds and therefore also increase the tendency to build nitrosamine-contaminated foods [57]. Adversely, through a PPA process, nitrite concentration is also increased. Nitrite might develop from the reduction of nitrate, but it is also a compound that accumulates through PPA gas admission. Additionally, the concentration of nitrous compounds may alter through extensive storage times. For instance, data have been published on the nitrite and nitrate content of vegetables, milk and dairy products, and other basic diary constituents. Nitrate has been shown to increase during the storage of raw vegetables with high nitrite content [58,59,60]. Many canned foods contain nitrate that formed under conditions of sterilization and of corrosion [61]. Against that background, nitrate and nitrite seem to be ubiquitous compounds that possess the ability to develop toxicity not only whilst in production but also during food storage. A PPA treatment may adversely enhance that process. On the other hand, the PPA source and its pronounced controllability opens the possibility for product-dependent decontamination processes that combat microbiological contamination without any excessive nitrosamine production. Although a PPA treatment might only be effective to a limited extent and only raw salmon reacts with a shelf-life prolongation, it may be used to support curing methods such as smoking.

However, contrary to curing smoke, PPA mixtures predominantly consist of nitrous gases that further react with the foodstuff’s moist surface to form nitric and nitrous acids. It has been found that moisture is a crucial precondition for a surficial counteraction on the microflora of any foodstuff [20,27]. Both curing smoke and PPA are complex chemical mixtures. Whereas curing smoke contains high and low molecular weight compounds that form during pyrolysis, they are more or less water-soluble. This is of great importance in the production of liquid smokes, because water-soluble fractions are enriched with constituents desirable for the smoke treatment of foods, while the water-insoluble fraction contains tar, solid particles (e.g., soot), and polycyclic hydrocarbons (PAH), some of which are very potent cariogenic compounds. The PPA provides a very comparable picture with a large number of identified or, in parts metastable, unidentified compounds. When a low temperature plasma processes air, a multiplicity of different biological active agents is produced in dependence on the adjusted parameters such as gas composition, flow rate, moisture, and temperature [25]. The PPA consists of a variety of potent compounds and plasma-triggered reactions that counteract the microflora that grows on almost every foodstuff surface. The spectrum of active compounds includes chemical products such as N_x_O_x_, atomic oxygen (O), ozone (O_3_), hydroxyl (OH), reactive oxygen (ROS), and nitrogen species (RNS), or larger charged particles. In addition, it also includes high-energy UV radiation, radiation in the visible and in the infrared spectral range and alternating electric fields; various regions of heat or physical and chemical etch processes may also occur and determine the properties of a plasma. In our experiments, UV radiation is largely irrelevant, as sample treatment and PPA production take place locally separately. However, especially a mixture of different agents makes a plasma attractive, because it is almost impossible for pathogens to develop resistance against these different kinds of plasma stress factors.

Furthermore, no synergetic effect was found between the PPA treatment and the cold-smoking process, which is, e.g., described for a combined treatment with PPA and plasma-treated water. Specifically, Schnabel et al. describe a synergistic effect for a combined treatment of PPA and PTW. The combined treatment reveals a reduction of bacteria that exceeds the sum of the individual possible reductions of the treatment types [62,63]. Unlike a PPA-PTW system, which has many chemical compounds in common, or PTW reactions that are based on PPA compounds, the composition of PPA and the pyrolytic wood smoke are very different and only share a similarity in controlling bacteria or pathogens. In a PPA wood smoke system, compounds might form that neutralize or silence each other. For instance, polyphenols are one major compound in the cold smoke that act as a decontaminant [21,22,64]. Polyphenols, as an anti-oxidant, might consume the oxidizing nitrous gasses, which are, on the other hand, the active compound in PPA [65,66]. Further investigations, which combine a PPA treatment and a cold-smoking process as consecutive steps, have not been conducted in the framework of the presented experiments, but need to be carried out for a reliable statement.

Beside the more conventional curing methods, irradiation [66], pressure [67], and active packaging [68,69,70] are methods that are frequently used in an industrial production process. Since these methods appear very exotic for consumers, a PPA treatment is analogous to an ordinary curing method that might be advantageous for consumer acceptance. However, all methods promote food safety and they are frequently used with great success. For instance, a treatment with a 10 eV electron beam reveals very comparable results where the matrix of the samples does not change due to the treatment. In contrast, the authors describe changed parameters such as the a*-value of the salmon samples that have been affected by irradiation dose dependency [71]. The tvc found on the samples was also significantly affected by different doses, which were monitored via a bacterial change in the total volatile basic nitrogen [72]. An additional method applied for shelf-life prolongation and an enhancement of food´s security is the use of high pressure. Erkan and co-authors [65] describe a shelf-life prolongation of up to two weeks after a high-pressure treatment without any significant changes in quality parameters. Lakshmana and co-authors additionally describe a lower total viable count of L. monocytogenes after a high-pressure treatment [73]. Generally, a high-pressure treatment appears a promising method for the improvement of the physicochemical, microbial, and sensory quality of fish muscle tissue [74,75]. As impact markers for an E-beam or a high-pressure treatment, most of the mentioned publications include further quality measures such as the affection of physiological pathways by a treatment or altered concentrations of compounds as its outcomes. Particularly, further investigations would allow us to definitely classify PPA treatment among existing methods. However, a PPA treatment solely alters the color perception when over-treated and it leaves the fish tissue completely unchanged in terms of color and texture in cases of mild treatment. In contrast, we see no effect of a PPA treatment when css is treated and no increase in shelf life is evident. Additionally, a decrease in the tvc, which was only observed in the treated rs samples, does not lead to an increase in shelf life of more than two days.

## 5. Conclusions

All experiments indicated that a PPA treatment significantly lowers the microbial load on raw salmon. Such a treatment prolongs shelf life by approx. two days. Therefore, the treatment divides into two process steps that house a PPA-generating pre-treatment (PT) and a post-treatment (POT) that govern the antimicrobial potential of the PPA. However, below a PT of 15 s, which generates a PPA with a moderate reactivity, a POT up to 5 min lowers the bacterial load on the rs samples without harming the fish matrix in terms of color or texture. Thus, a PPA process might be a meaningful support for conventional cold-smoking processes or the treatment of freshly slaughtered fish. It promotes food safety and prevents high food losses.

## Figures and Tables

**Figure 1 foods-11-03356-f001:**
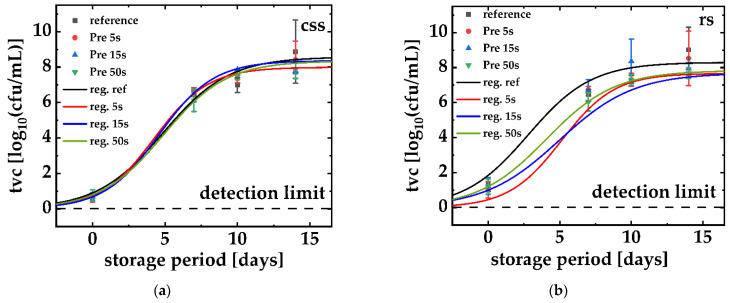
PPA-treated (**b**) and cold-smoked (**a**) salmon. The cold-smoked salmon underwent a 14-day storage period (day 0 until day 14) with sample days at day 0, 7, 10, and 14. The taken samples underwent a total viable count (tvc) of cultivable microorganisms that are hosted in their natural microbiome. Each data point mirrors the arithmetic mean (*N* = 5) and its standard deviation.

**Figure 2 foods-11-03356-f002:**
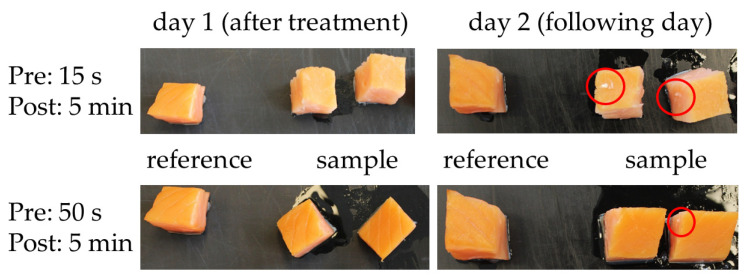
Photographs of cold-smoked salmon samples directly and 24 h after a PPA treatment. The upper line shows samples that have been treated with a set of optimum treatment parameters, the lower line shows samples that have been PPA-over-treated. The red circles depict whitish flocculation found on the salmon surfaces after a PPA treatment.

**Figure 3 foods-11-03356-f003:**
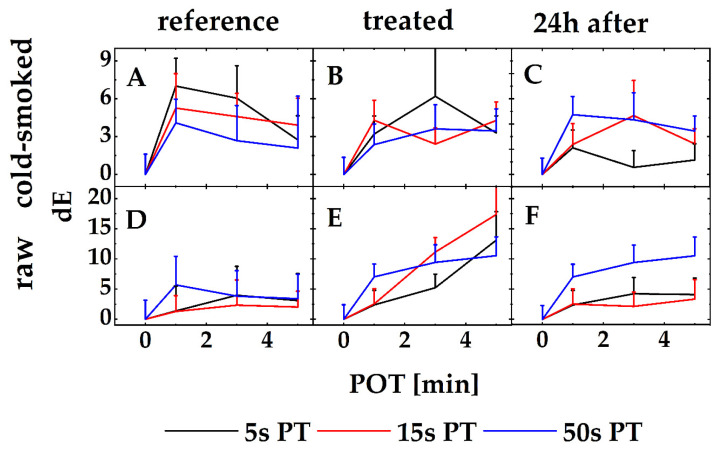
Color stimulus specification of raw and cold-smoked salmon samples that underwent a PPA treatment. All sample means were compared to an untreated reference, which mirrored naturally occurring color stimulus variations when carried in parallel to the experimental PPA treatment (**A**,**D**). (**B**,**E**) depict PPA-treated css (**B**) and rs (**E**). (**C**,**F**) reflect the same samples for css (**C**) and rs (**F**) after a storage period of 24h. All curve arrays start at a POT of 0 s, which is equivalent to an untreated reference used in the curve arrays of (**A**,**D**). Each data point mirrors the median (*N* = 3) and its error bars.

**Figure 4 foods-11-03356-f004:**
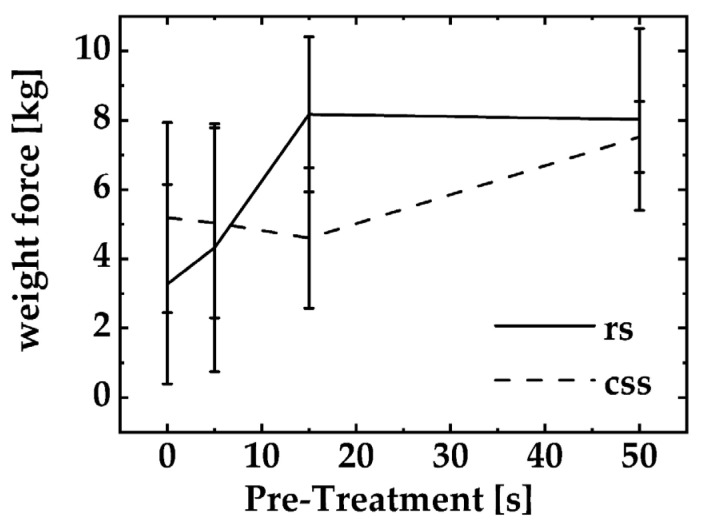
Texture specification of rs (solid) and css (dashed) after a PPA-treatment. The treated samples have been compared to an untreated reference of raw or cold-smoked salmon. Each data point mirrors the arithmetic mean (*N* = 5) and its standard deviation.

## Data Availability

The data that support our findings are available for download at https://doi:10.5281/zenodo.7113191.

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
