# Peer review of "Microbial Control of Raw and Cold-Smoked Atlantic Salmon (Salmo salar) through a Microwave Plasma Treatment"

_foods, 2022, doi:10.3390/foods11213356_

Round 1

Reviewer 1 Report

The authors present an article focused on the antimicrobial activity of microwave-plasma treatment. The topic is interesting because it provides a useful and important information regarding the applicability in the agri-food industries of a technology as widespread as plasma treatment.  The market and the consumption of the raw and smoked salmon is worldwide, and the hazards contained in these products could be many. Among the physical, chemical and biological hazards, pathogenic bacteria responsible for foodborne disease occupy an important place. This research could be used as a small-scale pilot study that could easily be transferred to industries, fueling and strengthening the relationship between the scientific communities and the industries.  

In my opinion, the manuscript is well-written and accurate; each aspect is taken into account. Some minor revisions are suggested and listed below.

I would like to suggest the authors to use also in the introduction section the whole words and put the abbreviations in round brackets (for examples, in the case of “css”). 

Please, check the scientific name of the bacteria (italics).

In the section Materials and Methods only the numbers of samples are missing and is not clear if the rheological analysis (paragraph “quality Control”) has been carried out during the whole shelf-life or only to compare the samples before and after the plasma treatment. Please, could you specify? 

Please, could you also specify the agar medium used for the microbiological analysis?

In the results section,  the authors referred to Table 1 and Table 2 to show the results of β-values, but tables in the text are not present. 

Could you please explain why only tvc was considered? Also, in M&M you referred to tvc at 33 and 37 ° C, but in the results section there is no mention of those.

Please, explain in what the three lines in figure 3a and 3d differ if they referred to reference samples.

Line 321, please check the number of the figure.

Please, among the other technics capable to extend the shelf life of raw salmon you should cite the active packaging (coating of antimicrobial peptides).

As far as the discussion, I would suggest the authors to reduce same parts, in which the focus is not their treatment. 

Overall, the text needs a moderate English check. There are some typos and grammatical errors, especially in the discussion part. 

Reviewer 2 Report

This manuscript studied the microbial control of raw and cold-smoked Atlantic salmon (Salmo salar) by microwave-plasma treatment. Although the structure is suitable, it still needs to be greatly modified.

- Line 41 Write (Ready-to-eat) as (ready-to-eat).

- Line 65 Provide a reference to the sentence “Due to an incomplete burning of the wood…”

-First, why is there no statistically significant difference between the untreated group and the untreated control group when using css. Or what is the significance of plasma-processed air processing? In addition, PT did not improve the bactericidal ability during rs treatment for a long time.

- Many units in the article are not uniform, such as Line 151 and Line 264 (Figure 1).

- The changes of processed samples can be circled in the picture (Figure 2).

- The results of Figure 3 B and 3 C are not explained, or some writing errors occur, such as Line 321.

- Line 420 and Line 421 Authors should apply new nomenclature for Lactobacillus species, as it was divided into 25 genera. E.g. Lactobacillus farciminis is now Companilactobacillus farciminis. See: LACTOTAX webpage http://lactotax.embl.de/wuyts/lactotax/

-Line 424 Listeria monocytogenes is not abbreviated.

- Line 432 Be careful to describe acronyms in full at the first mention (e.g., RONS).

-Check the capitalization of the reference title.

- Less than 1/2 of the references in recent five years.

Round 2

Reviewer 2 Report

No more questions and suggest to accept it.